# Evaluation of the Use of Digital Mental Health Platforms and Interventions: Scoping Review

**DOI:** 10.3390/ijerph20010362

**Published:** 2022-12-26

**Authors:** Luke Balcombe, Diego De Leo

**Affiliations:** Australian Institute for Suicide Research and Prevention, School of Applied Psychology, Griffith University, Messines Ridge Road, Mount Gravatt, QLD 4122, Australia

**Keywords:** mental health care, suicide prevention, digital mental health platforms, digital mental health interventions, evaluation, targeted strategies

## Abstract

Background: The increasing use of digital mental health (DMH) platforms and digital mental health interventions (DMHIs) is hindered by uncertainty over effectiveness, quality and usability. There is a need to identify the types of available evidence in this domain. Aim: This study is a scoping review identifying evaluation of the (1) DMH platform/s used; and (2) DMHI/s applied on the DMH platform/s. Methods: The Preferred Reporting Items for Systematic Reviews and Meta-Analyses extension for Scoping Reviews (PRISMA-ScR) guided the review process. Empirical studies that focused on evaluation of the use and application of DMH platforms were included from journal articles (published 2012–2022). A literature search was conducted using four electronic databases (Scopus, ScienceDirect, Sage and ACM Digital Library) and two search engines (PubMed and Google Scholar). Results: A total of 6874 nonduplicate records were identified, of which 144 were analyzed and 22 met the inclusion criteria. The review included general/unspecified mental health and/or suicidality indications (*n* = 9, 40.9%), followed by depression (*n* = 5, 22.7%), psychosis (*n* = 3, 13.6%), anxiety and depression (*n* = 2, 9.1%), as well as anxiety, depression and suicidality (*n* = 1, 4.5%), loneliness (*n* = 1, 4.5%), and addiction (*n* = 1, 4.5%). There were 11 qualitative studies (50%), 8 quantitative studies (36.4%), and 3 mixed-methods studies (*n* = 3, 13.6%). The results contained 11 studies that evaluated the DMH platform/s and 11 studies that evaluated the DMHI/s. The studies focused on feasibility, usability, engagement, acceptability and effectiveness. There was a small amount of significant evidence (1 in each 11), notably the (cost-)effectiveness of a DMHI with significant long-term impact on anxiety and depression in adults. Conclusion: The empirical research demonstrates the feasibility of DMH platforms and DMHIs. To date, there is mostly heterogeneous, preliminary evidence for their effectiveness, quality and usability. However, a scalable DMHI reported effectiveness in treating adults’ anxiety and depression. The scope of effectiveness may be widened through targeted strategies, for example by engaging independent young people.

## 1. Introduction

### 1.1. Background

Mental illness and suicide are ongoing primary global health problems [1] that need accessible and scalable solutions. For example, digital mental health (DMH), which is a contemporary method of mental health care that is distinguished by the large-scale integration of telehealth [2], apps [3,4], and digital platforms [5] as well as the promise of big data, genomics and artificial intelligence (AI) [6]. DMH platforms are a key technology for the purpose of assessment, support, prevention, and treatment in mental health [7]. Generally, digital platforms are an online space to exchange products, services, and information. The DMH global market is predicted to grow from USD 2568.6 million in 2021 to USD 18,717.5 million by 2030, at a compound annual growth rate of 21.1% [8]. An overview of systematic reviews summarized the research on the effectiveness of technology in DMH and found an extensive amount of DMH interventions (DMHIs) to address gaps in mental health service provision, in addition to shifting focus and target populations [9]. A hindering issue for the advancement of DMH is the sustained engagement of service users [10]. Therefore, it is important to provide a systematic approach to discern which DMH platforms and DMHIs are effective, usable and of good quality. Furthermore, it is necessary to clarify what mental health indications and populations these digital solutions are suitable for. To our knowledge, there is a lack of reviews that identify the types of available evidence on the use of DMH platforms. 

The aim of this scoping review is to describe:1)Empirical studies that focused on evaluation of the DMH platform/s used; and2)Empirical studies that focused on evaluation of the DMHI/s applied on the DMH platform/s.

To this end, we first provide an overview of existing work on (1) The use and functionality of DMH platforms, (2) Effectiveness of and engagement with DMHIs, (3) Implementation barriers for DMH platforms, (4) Recommendations for overcoming implementation barriers, (5) Evaluative research for the use of DMH platforms and DMHIs, and (6) Convergence of empirical and theoretical literature to increase effectiveness of DMHIs.

### 1.2. Overview of Existing Work

1)The use and functionality of DMH platforms 

Digital platforms are used in various contexts in DMH (see Section A.1 for definitions). For example, DMH platforms are used in more than 100 services for adults with anxiety and depression [11,12]. There is a priority to establish evidence for use in servicing people with diagnosed mental disorders [5]. DMH platforms are also used to assist early intervention strategies for young people. For example, to assist practitioners to deliver quality, personalized and measurement-based care for young people’s overall health, mental health, everyday function, suicidal thoughts/behaviors and social connectedness [13]. The use of digital platforms for video chats, social networks, telephone calls, and emails as a means of communication are effective at the population level for anxiety and depression although screening and intervention, AI-driven technologies, social media and digital phenotyping are generally not effectively used in DMH [14]. 

Internet-delivered cognitive behavioral therapy (ICBT) is the most used DMHI. ICBT is widely accessible, efficient, (cost-)effective and adaptable [15,16]. Self-guided treatment (28.4%) and guided telehealth/peer-to-peer approaches (16.3%) are the most used DMH services followed by real-time AI diagnostic assessments in computational psychiatry (13.7%), consumer journaling and support signposting (10%), physical, augmented and virtual reality (6.8%), diagnostic support (6.3%), gamified digital treatments (5.3%), neurological interventions (4.7%), digital phenotyping (4.2%), and virtual assistants (4.2%) [17]. Suicide prevention standalone digital platforms are rare because they are usually combined with DMH platforms [18]. 

The different types and uses of DMH platforms means it is necessary to distinguish among them in terms of functionality, which is its usefulness, or how well it performs the designated job. For example, the futility of risk assessment in psychiatry means the functionality of AI based DMH platforms is dependent on it being combined with personalized mental health care [19].

2)Effectiveness of and engagement with DMHIs 

A systemic review found several efficacious, scalable and sustainable suicide prevention interventions providing the opportunity for population-level impact and strategies to enhance effectiveness and reach [20]. Psychiatric diseases contribute to 60–98% of suicides [21]. Suicide prevention DMHIs may help augment ongoing clinical care if practitioners exercise caution in recommending suitable interventions and are aware of the security of the data that is collected [18]. Although integrating DMHIs into psychiatric care shows promising results for real-time monitoring and feedback on changes in common symptoms (e.g., stress, anxiety, and depression) [22], caution needs to be exercised in making recommendations for interventions on distress and suicidality because of uncertainty about their effectiveness and evaluation [19].

Meta-analyses of randomized controlled trials (RCTs), an experimental form of impact evaluation with a randomly selected sample and control group from the same population, noted potential efficacy for DMHIs for anxiety [23,24] and depression [25] in general populations. It was suggested to focus studies on comparisons with face-to-face psychological care [23]. This focus may help extract which aspects of the technologies produce beneficial effects and for which populations [25]. It may also help focus more studies with routine care populations [24]. There is a good potential for DMH platforms to be used in applying affordable interventions and preventive treatments [26,27,28]. However, the consumer marketplace is currently inundated with apps that lack engagement and efficacy [5]. Systematic reviews found a lack of clear and comprehensive evidence-base although there is a growing consensus that the most effective DMHIs are used for anxiety and depression particularly with college students [29] and young people [30]. A systematic review reported DMHIs have higher sustained engagement than self-guided digital tools [10]. This finding was endorsed by meta-analyses centered on anxiety [31] and depression [32].

3)Implementation barriers for DMH platforms 

A range of barriers hinder effective and sustained implementation of DMH platforms. For example, the field is constrained by issues of affordability [33], accessibility, relevance, reliability, a lack of personalization and human capacity [12], technical and ethical considerations [34] as well as privacy and security, efficacy, engagement, and clinical integration [5]. There is rigorous evidence of efficacy in trials although a lack of real-world impact [35] means there is an inconsistent impact. This is because of difficulties in instructing patients and mental health care professionals in using DMH platforms as well as the regulatory context of health care delivery [5]. The promising results in support of DMH platforms may be hindered by the human factors of human–computer interaction (HCI) (e.g., organizational readiness and usability in the healthcare context) [14]. For example, there was a 500% increase in the use of tailored self-guided resources by healthcare workers during the COVID-19 pandemic, although most dropped out of treatment because of time constraints, privacy concerns, treatment relevancy and satisfaction with the digital health platform design and experience [36].

4)Recommendations for overcoming implementation barriers 

Different levels of DMH platform evaluation are required ranging from feasibility and pilot studies on user retention/acceptability, safety and satisfaction through to RCTs and implementation feasibility studies [37]. Apps need to be moved to an integrated digital platform, and digital tools need to be highly effective and engaging, address inequalities, and build trust in their authenticity [35]. There also needs to be better (cost-)effectiveness [38,39]. Furthermore, innovation is required to converge pattern-based and hypothesis-driven methods for evaluation of rigorous preventive strategies and interventions [5,19,40]. Codesign may help to strengthen the human-centered design process and instill an understanding of how an application achieves real-world effectiveness [14]. All the aspects surrounding innovation must be considered for the sustained use of DMH platforms. ‘Convergence mental health’ is recommended to facilitate access to and use of DMH services through integrating scientists, clinicians, bioinformaticists, global health experts, engineers, technology entrepreneurs, medical educators, caregivers, and patients as well as infusing synergy between government, academia, and industry for multidisciplinary applied and translational solutions [41].

5)Evaluative research for the use of DMH platforms and DMHIs

There is a small amount of previous review and analysis on (1) evaluation of the use of DMH platforms and (2) evaluation of the use of DMHIs. As an example of 1, the DMH platform MOST was applied in evaluative research that highlighted the potential of novel multimodal approaches to help-seeking by connecting MOST with clinical services to provide support in real-time and to sustain mental health recovery for young people [42]. An earlier pilot study established the acceptability, safety and initial clinical benefits of the Horyzons DMH platform for peer-to-peer social networking, individually tailored interactive psychosocial interventions, and expert interdisciplinary and peer-moderation [43]. MOST was reported to be safe and effective for evidence-based mental health support for young people with psychoses, depression, social anxiety, mental illness and suicidal risk [44]. As an example of 2, an RCT study demonstrated the efficacy of an ICBT program—‘Space from Depression’—for adults with depressive symptoms [45]. 

6)Convergence of empirical and theoretical literature to increase effectiveness of DMHIs

An integrated blueprint suggested eminent DMH platforms are needed to increase the effectiveness of DMHIs in self-guided and guided approaches [46]. The lack of highly effective, evaluated DMH platforms is entrenched in the struggle to sustainably innovate. There are underlying quality, safety and usability issues stemming from the difficulty converging theoretical, data-driven/technological and empirical research, as well as to satisfy mental health care professionals’ and users’ HCI demands [19,47,48]. The development of optimized patient-centric digital tools is not the problem. Rather, it is how long it takes mental health care professionals to adapt in using these tools. For example, DMHIs may assist the prevention of the sequalae of mental illness quickly and accurately through predictive systems that apply DMH platforms and AI-driven apps [19,39,49,50]. A trial-and-error approach may be necessary to overhaul how codesign, behavior theories, and clinical evaluation are applied [51]. There is also a need to confront the lagging human factors that limit the successful implementation of DMH platforms and effective industry standards.

## 2. Methods

### 2.1. Overview

A scoping review methodology was undertaken to summarize empirical studies that evaluated web-based, smartphone and cross-platform DMH platforms and DMHIs used in assessment, support, prevention, and treatment for all indications of mental health disorders as well as suicidality. The reason for focusing on all mental health disorders (i.e., schizophrenia; anxiety, bipolar, depressive, autism spectrum, attention deficit hyperactivity, conduct and other mental disorders; idiopathic developmental intellectual disability; and eating disorders) is because prevention and early intervention are important for decreasing the mental illness sequalae and new ways of assessment, support and treatment may be possible with DMH [19]. Suicidality is included because it may sometimes occur separate from a mental health disorder. The methods selection was guided by the purpose and framework of scoping reviews [52] and the description of the 6 different exemplars for scoping reviews [53]. Exemplars provide an ideal model to follow. We chose to follow the exemplar ‘to identify the types of available evidence’. Our review started with planning the review procedure and continued with a search process and practical screening of articles to identify evidence. We focused on evaluation of the DMH platform/s used in addition to evidence focused on evaluation of the DMHI/s applied on the DMH platform/s. Issues with generalizability and validity were mostly unknown because of a large body of evidence in the domain. The intention was to enable knowledge by clarifying the type of DMH platform used in the empirical study and in what context it was evaluated. We described the type of study/aim, its purpose, population, outcomes/form of evidence as well as the model/s of care applied. The aim and outcomes of the study were examined to determine the types of DMH platforms and DMHIs used. 

Included studies were selected and assessed for compliance with predetermined inclusion criteria. These were described and illustrated according to a modified Preferred Reporting Items for Systematic Reviews and Meta-Analyses—extension for Scoping Reviews (PRISMA-ScR) [54]. This procedure identified the current position of the evidence in the domain by separating studies that evaluated the use of DMH platform/s from those that evaluated their use in DMHI/s. Studies that primarily used apps, AI-driven immersive/interactive/wearable technologies, social media and digital phenotyping for mental health care and/or suicide prevention were out of scope because digital platforms are the most used technologies in self-guided and guided approaches [17]. Therefore, digital platforms are the most likely technology to be associated with evidence. There are safety and quality concerns about apps because there are more than 10,000 available [55], and apps have a low rate of testing (30%) for individuals with clinical conditions [56]. 

### 2.2. Search Strategy

Based on the research aims, the search terminology “digital platform” AND “mental health care” OR “suicide prevention” was used on 7 April 2022 to search full text journal articles in 4 databases—Scopus, ScienceDirect, Sage, and the Association for Computing Machinery (ACM) Digital Library. The same search terminology was used in 2 search engines (PubMed and Google Scholar). A combination of other search terms was tested. These databases and search engines were tested for variance in searches of the following search terms “digital mental health”, “platform”, “multifunctional”, “mental health care”, “distress”, “suicide prevention”, “suicide behavior prediction”, “self-help”, “guided”, “digital interventions”, “depression”, “anxiety”, “suicide” and “wellbeing”. However, the results of various combinations of these search terms found no further relevant articles. Therefore, the other search terms were excluded, and 22 articles were deemed to be suitable for inclusion, underlining the narrow focus of the field. 

The ACM database was selected to cover computing and information technology articles. PubMed was selected to include medical and psychology-related articles. Scopus, ScienceDirect, Sage and Google Scholar were chosen to include studies in multidisciplinary areas of interest including psychology, the social sciences, and hybrid studies that used digital platforms in the study. Article types included qualitative, quantitative and mixed-method studies published between 2012–2022. We distinguished between qualitative and quantitative studies (including clinical trials) from the following health-focused research methods definition by Denny and Weckesser. Qualitative study designs are characterized by aiming to provide insight and understanding of an individual’s experience in terms of thoughts and behaviors, whereas quantitative research aims to detail what happened, for example through applying randomized evaluations [57]. The title, abstract, keywords were screened. All articles were in English. The evidence-base was inferred to be mostly from within the previous 5 years although the inclusion criteria was increased to the prior 10 years to minimize selection bias. For example, the systematic overview on evidence for DMHIs for young people by Lehtimaki et al. [30] applied a prior 10-year period in the inclusion criteria although the 4 systematic reviews included were from the prior 4 years.

The inclusion and exclusion criteria and data extraction format were drafted by the first author (LB) and then reviewed and finalized in coordination with the co-author (DDL). The preliminary search process involved a screening of the search results carried out by the first author. Data extraction and full-text review were performed by the first author applying the inclusion and exclusion criteria. A quality appraisal and consultation with the co-author was applied to reduce bias and uncertainty and to create reliability and trust in the research. Ambiguities were reduced through discussion and consensus among the authors. 

### 2.3. Inclusion and Exclusion Criteria

The inclusion and exclusion criteria (see Section A.2) informed the selection of studies. An article was kept if it met the inclusion criteria and was disqualified if it met any of the exclusion criteria.

### 2.4. Data Analysis and Synthesis

The first author extracted the data from the shortlisted articles, based on the research aims. The study design/aim, DMH platform (type, purpose of use, and population), outcomes/form of evidence and the approach/comparison were tabled to organize the evaluation. A table organized studies that focused on the DMH platform/s in addition to a table on the DMHI/s applied on the DMH platform/s. If these details were not clear, the DMH platform or DMHI was analyzed for what it was put in place for. Therefore, each study’s aim was compared with its outcomes to determine if it was primarily focused on evaluation of the DMH platform or the DMHIs applied on it. If the population details were not clearly stated, then the user recipients were extrapolated through interpretation. For example, it was inferred that the Swedish general population use the Swedish health care system. The DMH platforms were categorized according to our previous reviews that identified tele-mental health, online self-guided and/or online guided therapy, as well as multifunctional and/or integrated DMH platforms as the 5 main types reported [47,48]. Due to the heterogeneity of the included studies, a narrative synthesis was undertaken. 

## 3. Results

### 3.1. Selection of Articles

In total, 6879 records were retrieved from databases and search engines including: 3346 (48.6%) from ScienceDirect (11 were assessed for eligibility and 11 were excluded); 1481 (21.5%) from PubMed (1 was assessed for eligibility and it was excluded); 1010 (14.7%) from Google Scholar (75 were assessed for eligibility—7 were included and 68 excluded); 804 (11.7%) from Sage (12 were assessed for eligibility—2 were included and 10 excluded); 145 (2.1%) from Scopus (25 studies were assessed for eligibility—9 were included and 16 excluded); 75 (1.1%) from ACM Digital Library (2 studies assessed for eligibility and 2 were excluded), and 18 (0.3%) records from additional sources (i.e., reference lists of included studies—18 studies assessed for eligibility—13 were excluded and 5 included). 

Out of the 6879 records retrieved, 5 duplicates were removed. Therefore, 6874 records were screened by reading their title, abstracts and keywords. Full texts of 144 records (2.1%) were assessed for eligibility—22 (15.3%) empirical studies met the inclusion criteria and 122 (84.7%) were excluded. The reasons for exclusion were because the articles were assessed to be about digital platforms with no mental health care or suicide prevention outcomes, descriptions of DMH platform development with no outcomes, DMH platform trial descriptions with no results, and follow up articles with the same DMH platform. Studies were checked for follow-ups with the same digital solution—1 article was excluded on this basis—the most recent and better-quality findings were included. The selection process (see Figure 1) was based on a modified version of the Preferred Reporting Items for Systematic Reviews and Meta-Analyses—extension for Scoping Reviews (PRISMA-ScR) [54]. See Appendix B for the PRISMA-ScR Checklist.

### 3.2. Summary of Results

The scoping review findings are summarized in two overviews. Firstly, for the 11 empirical studies that focused on evaluation of the DMH platform/s used (see Table 1). Secondly, for the 11 empirical studies that focused on evaluation of the DMHI/s applied on the DMH platform/s (see Table 2). 

#### 3.2.1. Main Characteristics of the Included Studies

The studies were conducted in Australia (*n* = 10, 45.5%), Europe (*n* = 6, 27.2%) and North America (*n* = 6, 27.2%).Most of the studies did not include specific age groups. It was inferred that 15 (68.2%) of the included studies were generally focused on adults and 7 (31.8%) of the included studies were focused on young people including children, adolescents, as well as college and university students aged 18–28.Most of the studies addressed the use of DMH platforms for general/unspecified mental health and/or suicidality indications (*n* = 9, 40.9%), followed by depression (*n* = 5, 22.7%), psychosis (*n* = 3, 13.6%), anxiety and depression (*n* = 2, 9.1%), as well as anxiety, depression and suicidality (*n* = 1, 4.5%), loneliness (*n* = 1, 4.5%), and addiction (*n* = 1, 4.5%).Targeted strategies were reported in 8/22 studies (36.4%) comprising of youth with psychosis (*n* = 3, 13.4%), depression and stress in LGBTQA+ youth (*n* = 1, 4.5%), secondary students with symptoms of anxiety and depression (*n* = 1, 4.5%), mothers with postpartum depression (*n* = 1, 4.5%), loneliness in adults (*n* = 1, 4.5%), and adults with addictions (*n* = 1, 4.5%).The types of DMH platforms used were integrated (*n* = 5, 22.7%), integrated-multifunctional (*n* = 5, 22.7%), guided therapy (*n* = 5, 22.7%), self-guided and guided therapy (*n* = 3, 13.6%), multimodal (*n* = 1, 4.5%); self-guided (*n* = 1, 4.5%), direct to consumer tele-mental health (*n* = 1, 4.5%), and an unspecified range of existing DMH platforms (*n* = 1, 4.5%).The studies were mostly investigated with a blended mental health care approach (*n* = 11, 50%). Some were combined with a comparison approach: blended mental health care and usual primary care (*n* = 2, 9.1%); blended mental health care and waitlist control (*n* = 2, 9.1%); blended mental health care and online self-guided (*n* = 1, 4.5%). Stepped mental health care approaches were less common and combined with comparisons where implemented: stepped mental health care and self-guided (*n* = 1, 4.5%) and stepped mental health care and waitlist control (*n* = 1, 4.5%). Other studies used self-guided approaches (*n* = 1, 4.5%) or self-guided and guided approaches (*n* = 3, 13.6%).Overall, there were slightly more qualitative studies (*n* = 11, 50%) than quantitative studies (*n* = 8, 36.4%) including 4 RCTs, in addition to a few mixed-methods studies (*n* = 3, 13.6%).Feasibility (*n* = 6, 27.25%) was the most common study type in addition to various combinations, i.e., feasibility and acceptability (*n* = 3, 13.6%); feasibility, acceptability and engagement (*n* = 2, 9.1%); feasibility, usability and engagement (*n* = 1, 4.5%); and feasibility, safety and acceptability (*n* = 1, 4.5%). The remainder of the study types included usability and engagement (*n* = 4, 18.2%); effectiveness (*n* = 2, 9.1%); effectiveness and usability (*n* = 1, 4.5%); acceptability (*n* = 1, 4.5%); and acceptability and engagement (*n* = 1, 4.5%).

#### 3.2.2. Main Findings of the Included Studies

A review of the empirical literature found a small but promising amount of evidence for the use of DMH platforms and DMHIs in mental health care and suicide prevention. Overall, significant evidence was found in 2 of the 22 (9.1%) included studies. There was mostly preliminary evidence marked by 19 of the 22 (86.4%). No evidence was found in 1 of the 22 (4.5%). The 11 empirical studies that focused on evaluation of the DMH platform/s used were comprised of 1 study (9.1%) that found significant evidence, 9 studies (81.8%) that found preliminary evidence and 1 study (9.1%) that found no evidence. The 11 empirical studies that focused on evaluation of DMHI/s applied on the DMH platform/s were comprised of 1 study (9.1%) that found significant evidence and 10 studies (90.1%) that found preliminary evidence.


**Empirical studies that focused on evaluation of the DMH platform/s used**


One study provided significant evidence:One quantitative study on feasibility and acceptability found efficacy in the affirmative CBT-based AFFIRM Online which used blended care to relieve depression and coping with stress in the LGBTQA+ youth community [61].

Nine studies contributed preliminary evidence:One RCT on feasibility, acceptability and safety found Horyzons had no significant effect on social functioning compared with treatment as usual [58]. Although there was a significant correlation between the use of the DMH platform and perceived helpfulness for vocational and relapse prevention support.One quantitative study on feasibility and acceptability found statistically significant support for 7Cups in treating postpartum depression [59]. However, there was no significant difference compared to treatment as usual.One qualitative study on feasibility found Happify Health’s loneliness interventions may be effective in self-guided and guided approaches [60].One RCT on feasibility found possible efficacy for the Swedish health care system DMH platform that applies ICBT for treating depression in routine psychiatric care [62]. Although findings are limited by the small sample size.One qualitative study on
feasibility found stakeholders supported the use of the Innowell DMH platform [63]. Although effective implementation is hindered by human factors.One quantitative study on effectiveness found BetterHelp to be potentially effective for treating adult depression [64]. Although, it was noted that trials are needed.One mixed-methods study on feasibility, acceptability and engagement found initial support for Smooth Sailing [65]. Although, effective engagement strategies are needed.One qualitative study on usability and engagement found longitudinal studies are required to confirm Depression Connect is effective for sharing coping experience [66].One mixed-methods study on feasibility found DMH platforms can assist evaluating youth wellbeing [68]. However, more effective qualitative strategies are required.

One study demonstrated no findings of evidence:One qualitative study on acceptability and engagement found a lack of support for Virtual Coach because it was difficult to relate to and engage with [67].


**Empirical studies that focused on evaluation of DMHI/s applied on the DMH platform/s**


One study provided significant evidence:One RCT on the effectiveness of the SilverCloud DMH platform’s ICBT in a stepped care approach reported (cost-)effectiveness with significant long-term impact on anxiety and depression in UK general population adults [76].

Ten studies contributed preliminary evidence:One qualitative study on feasibility and acceptability found largely positive views on DMHIs for health care delivery [69]. However, concerns over privacy and data were noted.One qualitative study on
feasibility, usability and engagement reported user engagement and delivery of ICBT for depression could be improved by establishing, planning and promoting a working alliance in the user-practitioner relationship [70].One qualitative study on usability and engagement found tele-mental health on DMH platforms may offer a range of important interpersonal interaction that presents benefits [71]. Although, there are hindering ethical complexities and structural challenges.One qualitative study on feasibility
found SMART Recovery could assist mutual support through meetings online [72]. However, these methods are not as well-suited to those with experience of in-person support.One RCT on usability and engagement found an optimized UI based on UX contributed to increased usability and engagement in treatment with the Swedish health care system DMH platform
[73]. Although, the relationship between UI and treatment effectiveness was unclear.One qualitative study on feasibility found clinicians use digital tools with utility [74]. Although, a centralized DMH platform is required to improve stakeholder accessibility in addition to youth-oriented tailored solutions.One mixed-methods study on feasibility, acceptability and engagement found consensus on the stakeholder benefits from DMHIs that use technology-enabled care coordination (TECC) [75]. However, implementation of the DMHIs is hindered by human factors.One qualitative study on usability and engagement found appropriate language and presentation styles in a social media campaign and online support forum [77]. However, datasets are required to improve mental health communication.One quantitative study on effectiveness, usability and engagement found a high level of engagement and a very high level of satisfaction and sustained overall improvement in psychological symptoms [78]. Although, the relatively small size of the registered sample prevented generalizability.One qualitative study on acceptability found young people supported blended mental health care in an assistive capacity to traditional care although evaluative evidence is needed to determine the impact on the therapeutic alliance, clinical and social outcomes, cost-effectiveness, and engagement [79].

The most described services were mental health screening, online guided and online self-guided, tele-mental health, and integrated approaches. There were more studies on adults (68.2% compared to 31.8% for youth) although targeted strategies were more common for youth (62.5% compared to 37.5% for adults). Only a few studies focused on subpopulations, of which youth with psychosis was the most studied. However, there was efficacy found for AFFIRM Online which demonstrated a successful example of community based DMH platform for LGBTQA+ youth with stress and depression. Overall, an RCT with SilverCloud’s ICBT program was the most significant evidence to-date. Richards et al. noted the (cost-)effectiveness of a DMHI with significant long-term impact on anxiety and depression in the UK general adult population [76].

## 4. Discussion

### 4.1. Principal Findings of Empirical Literature

A slightly higher qualitative evidence base was found in comparison to quantitative studies although the difference was made up of mixed-methods studies. Overall, the studies mainly evaluated feasibility, usability, engagement, acceptability and effectiveness. Although feasibility was found for the use of DMH platforms and DMHIs in mental health care and suicide prevention, the results highlight the need to increase usability and engagement in addition to effectiveness and quality.

The main types of DMH platforms used in the 22 included empirical studies are categorized as integrated, guided, self-guided, integrated-multifunctional, multimodal, and direct to consumer tele-mental health. This contrasted with previous reviews which mostly reported off-the-shelf solutions through computers, mobile apps, text message, telephone, web, CD-ROM, and video for general population DMHIs for suicidal ideation and mental health co-morbidities [20]. Other previous reviews focused on general mental health support [10], in addition to self-guided digital tools for anxiety and depression in general populations [31,32]. In line with the previous reports of variability in the applications of use, the empirical evidence suggests DMH platforms and DMHIs are used for a range of purposes, e.g., to treat loneliness and to aid suicide prevention.

The high number and frequent use of DMH tools [9] is reflected in the evaluative evidence base on the use of DMH platforms and DMHIs. In line with the previous findings of Borghouts et al. [10], there was heterogeneity found in the mostly preliminary evidence. These findings mainly focused on feasibility, usability, engagement, and acceptability rather than the effectiveness of each DMH platform or DMHI.

The most significant finding overall arose from the RCT for SilverCloud’s ICBT for anxiety and depression [76]. This RCT proceeded a study that established efficacy with regards to ICBT for adults with depressive symptoms [45]. The general lack of study follow-up in the domain has hindered the evaluation when considering there are more than 100 DMH programs for depressed and anxious adults [11,12]. RCTs are considered the “gold standard” by which psychological interventions are evaluated and subsequently adopted into general clinical practice [80]. However, there are some limitations of RCTs in developing treatment guidelines in terms of the pragmatic application from a sample to the individual patient. For example, the baseline characteristics of the RCT by Richards et al. [76] reported that 70% of the sample were female, noting this is only slightly higher than program referral rate for females (65%). This incidental finding highlights the inherent difficulties in recruiting and engaging men in mental health research [48]. This limitation extends to identifying the underserved and the unserved in mental health care assessment and treatment [19].

The significant and preliminary evidence categories presented in the Results section do not tell the whole story regarding efficacy and effectiveness. For example, a previous review reported the DMH platform MOST is safe and effective [44]. However, it is not clearly stated what it is effective for. It appears from the qualitative study by Valentine et al. [79] that young people supported blended care through Horyzons (a derivative of MOST). Although, further evaluative research is needed on efficacy, e.g., on the therapeutic alliance, clinical and social outcomes, cost-effectiveness, and engagement. There was also no significant effect on social functioning compared with treatment as usual as a primary outcome of the RCT with Horyzons [58]. This RCT followed extensive design, implementation [42], and augmentation of social connectedness and empowerment in youth first-episode psychosis [43]. These examples highlight the need for the current study which distinguished between evaluative research focused on the effectiveness of the DMH platform as well as the effectiveness of the DMHI applied on the DMH platform.

The previous body of knowledge noted the difference between rigorous evidence of efficacy in trials and outcomes that indicate a lack of real-world impact [35]. The current study supports this finding. Although, it may help to also clarify about efficacy and effectiveness to generally assist in the evaluation of DMH platforms and DMHIs. For example, Craig et al. [61] evaluated the AFFIRM Online DMH platform and reported it brought about efficacy through working under ideal circumstances. However, the RCT for SilverCloud’s ICBT for anxiety and depression [76] was deemed to be more significant in evidence because it applied a waiting list to demonstrate pragmatic effectiveness by working in substandard circumstances. Evaluation of DMHIs may produce relevant, measurable, responsive, and resourced indications on safety or effectiveness for its intended mental health care and/or suicide prevention purpose. RCTs can bolster these claims by providing randomization which decreases bias and offers a rigorous tool to examine cause-effect relationships between an intervention and outcome. However, a successful RCT may not be required to demonstrate safety and effectiveness.

### 4.2. Secondary Findings of Empirical Literature

Robust stakeholder engagement is required to ensure there is responsiveness to needs and to gain support for DMH implementation. The previous review noted the existence of targeted strategies to serve young people in mental health care [13,42]. Although the evidence synthesis found more of a focus on adults, there was a slightly higher number of targeted strategies for young people. However, there is a need for more effective qualitative strategies such as in designing and implementing youth-oriented tailored solutions [68] and implementing a centralized DMH platform to improve stakeholder accessibility [74]. The previous review of Spadaro et al. [51] suggested overhauling the application of codesign, behavior theories, and clinical evaluation. In line, a qualitative study that evaluated DMHIs on the Innowell DMH platform articulated some implementation problems: restricted access, siloed services, interventions that are poorly matched to service users’ needs, underuse of personal outcome monitoring to track progress, exclusion of family and carers, and suboptimal experiences of care [75]. A consequential evaluation of the Innowell DMH platform led to the finding that national scalability is hindered by human factors—the main problem is not the technology but the humans that implement and use it [63]. This is in line with previous findings about the constraints in instructing the recipients of technologies [5] and transforming clinicians’ strong interest in using technology to actual use [4].

A previous review found human-centered design is important for the codesign process to instill an understanding of how DMH platforms can be used with engaging effectiveness [47]. However, human-centered design is often not implemented well in DMH services. The evidence for HCI issues was in line; for example, the relationship between the UI of a DMH platform and treatment effectiveness was unclear [73]. Furthermore, the results indicate that young people who perceived DMH platforms as useful in blended care were more willing to use the system in the future [69,79]. The results with the Innowell DMH platform suggested that codesign is not a foolproof method to increasing effectiveness with DMH platforms [63]. Previous findings on the need for key stakeholder and user input [3] were echoed in addition to the call for funding and resources to expand regional case studies to the state level and beyond.

### 4.3. Future Research Implications and Prospects

International collaborations were proposed for the Australian DMH platform MOST+ to be adapted, translated and developed in a digital transdiagnostic clinical–and peer-moderated treatment trial with youth in the Netherlands [81]. Although designed to serve adult Australians, MindSpot is a clinically validated, vetted DMH platform that has provided free psychological screening and ICBT treatment for anxiety, depression, mental well-being, and general distress to more than 500,000 users [78]. Successful engagement strategies were noted as required to increase the number of registered users to provide generalizability of the effectiveness of its ICBT program. However, there is a limit to the number of users that MindSpot can treat at a given time, so engagement strategies need to be tailored with this in mind. The most evidenced example of effectiveness is from an ICBT service for treating anxiety and depression in UK adults using the globally available, clinically validated SilverCloud DMH platform [76]. The organizational ability to increase registered engagement within the treatment capacity is an important issue the domain is grappling with.

Qualitative studies focused on increasing innovative engagement, usability and quality with adults suffering primarily from anxiety and depression may be a progressive next step to gather more evidence for the field. Then, it may be possible to translate findings and reevaluate (cost-)effective ways of targeting young people in mental health care and suicide prevention. It is already apparent from the preliminary evidence on serving young people that there are issues with DMH platform accessibility [74] and DMHI implementation [75]. Furthermore, there are issues in the need for parental consent/involvement, as well as higher uptake and engagement through frequent screening especially in adolescents [65].

The results were mostly derived from multidisciplinary databases—Scopus, ScienceDirect, and Sage. There is potential for future multidisciplinary research to focus on developing an understanding of what is technically required for an eminent DMH platform and how this can be applied with DMHIs. It can be surmised that an integrated-multifunctional DMH platform would be best used to demonstrate how to grow an ecosystem. For example, the LAMP DMH platform may be integrated with other systems and combines innovation, research and clinical interventions (e.g., assessment via surveys and sensors, digital phenotyping, self-management tools, data sharing with patients, and clinician support). LAMP is also linked to a consortium that provides education and collaboration between mental health practitioners and users to enable translational research [82].

The human factors problem noted in this review may benefit from a better understanding of interprofessional dynamics. An interprofessional approach appears necessary to promote mental health and well-being through various means including engagement, assessment, and intervention. It may help to investigate if the domain is encountering barriers that include competition among the various professionals, thus hindering effective outcomes. It could therefore be a prospect to incorporate a model of expertise-based care into the domain. For example, through combining interprofessional values and ethics, common and encompassing respect, in addition to privacy and confidentiality in service delivery.

## 5. Strengths and Limitations

This study is strengthened by a systematic approach applied to two different operationalizations of evaluating DMH platforms, i.e., the use of DMH platforms as well as how they are applied in DMHIs. The body of knowledge was synthesized to point towards the aims of the review. Next, the results were tabled and clearly presented. Thereafter, the discussion compared the previous body of knowledge with the results, integrating the complexities and challenges of evaluating DMH platforms and DMHIs as well as opportunities for increasing the evaluative impact of the domain. Furthermore, this review strengthens the knowledge base by clearly pointing to which types of DMH platforms and DMHIs as well as study designs may best help advance the domain.

Although a thorough effort was made to confirm the rigor of the search strategy, potentially appropriate studies may not have been identified if the authors of that journal article did not use the search keywords that were included in this review. For example, we may have missed articles that used alternate forms of “digital platform” AND “mental health care” OR “suicide prevention” such as “operating system” AND “psychological interventions” OR “crisis support”. It may help future systematic reviews to be consistent with the search terminology. Although, a wider range of alternate search terminology may be necessary as the domain advances. In addition, limiting the search to include journal articles published in the English language may also have excluded relevant studies in other languages.

A potential limitation is that we did not include studies focused on standalone mobile apps, AI-driven interactive/immersive/wearable technologies, social media and digital phenotyping. Bell et al. [4] included all these DMH technologies in their survey of young people’s and clinicians’ access to, use of and interest in various technologies and their applications. However, we focused on the evaluation of DMH platforms or DMHIs applied on DMH platforms. The rationale is based on the assessment of the body of knowledge that digital platforms are the most used technologies [17]. Although there is yet to be global eminence in the use of DMH platforms [46], it is established that there are a very large number of apps lacking in clinical testing [55,56]. The other excluded technologies are interesting. However, a line needed to be drawn regarding efficacy, safety and quality. The DMH platform types that we focused on (i.e., tele-mental health, self-guided and/or guided therapy, as well as multifunctional and/or integrated) were also described by Bell et al. [4]. Although, some of the terminologies were slightly different.

The review is compliant with PRISMA-ScR and included a quality appraisal, although there was no critical appraisal of individual sources of evidence because this was not in the research aim. It may be appropriate for future research to include PRISMA-ScR Item 12 ‘critical appraisal of individual sources of evidence’ including a rationale, a description of the methods used and how this information was used in any data synthesis.

## 6. Conclusions

This scoping review evaluated the varied use of DMH platforms and DMHIs for managing and assisting mental health care and suicide prevention. Although there is a need to decrease heterogeneity, and increase the number of significant findings, the review highlighted the promise of several usable, quality DMH platforms. A scalable DMH platform applying ICBT for treating adults’ anxiety and depression is currently the most reliable example of effectiveness. A notable challenge is implementing targeted strategies such as engaging independent young people.

## Figures and Tables

**Figure 1 ijerph-20-00362-f001:**
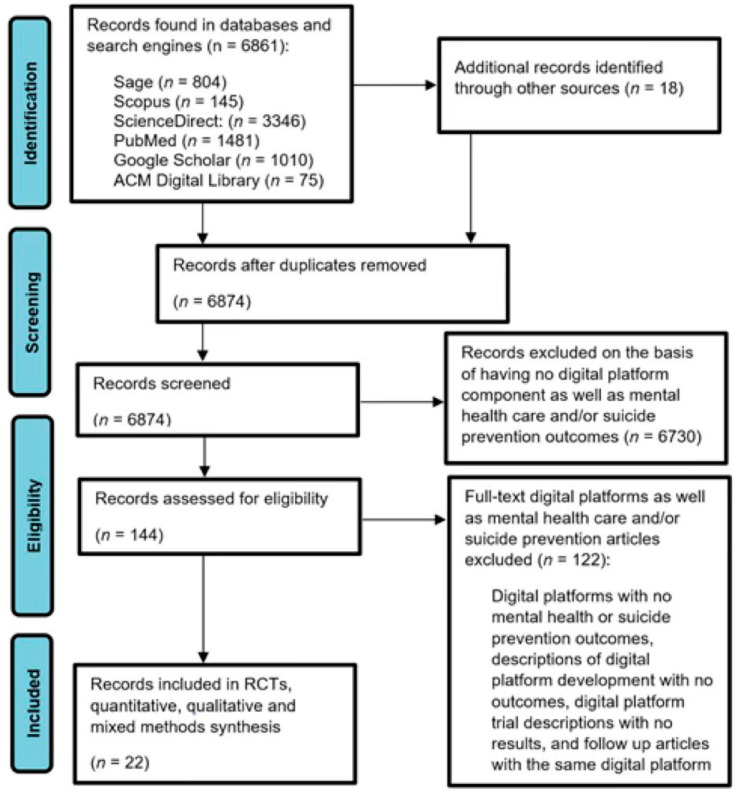
Flowchart of the study selection procedure.

**Table 1 ijerph-20-00362-t001:** Overview of empirical studies that focused on evaluation of the DMH platform/s used.

Ref-ere-nce	Authors	Study Design/Main Aim	DMH Platform (Type, Purpose of Use and Population)	Outcomes/Form of Evidence	Approach/Comparison
[58]	Alvarez-Jimenez et al. (2021)	RCT To ascertain the feasibility, acceptability, and safety of MOST+	Integrated-multifunctional DMH platform (Horyzons, a derivative of MOST)—used for targeting early intervention for youth psychosis (*n* = 170) through treatment, employment and education	Feasibility, acceptability and safety—no significant effect on social functioning compared with treatment as usual. Although there were significant correlations between system use, perceived helpfulness, and a number of secondary outcome variables, e.g., increased likelihood to enroll in education/find employment or less psychosis-related visits to hospitals and emergency services	Blended mental health care and usual primary care
[59]	Baumel et al. (2018)	Quantitative—survey (purposive sample) To examine the feasibility, acceptance, and preliminary clinical outcomes of using 7Cups	Self-guided and guided therapy DMH platform (7Cups)—online self-help tools and 24/7 emotional support delivered by trained volunteers—mothers with postpartum depression (*n* = 19) were targeted in an adjunct treatment	Feasibility and acceptability—7Cups significantly decreased postpartum depression treatment outcomes. Although there was no significant difference compared to treatment as usual	Blended mental health care and self-guided mental health care
[60]	Boucher et al. (2021)	Qualitative—focus group To explore how Happify Health may be an effective tool for disseminating loneliness interventions	Self-guided and guided therapy DMH platform (Happify Health)—used to target loneliness in adults aged 18–64 years (who indicated wanting to be more connected to others when signing up to the DMH platform) (*n* = 11)	Feasibility—preliminary evidence of effectiveness for using Happify Health in loneliness interventions. The DMH platform may be useful as a productive distraction	Self-guided and guided mental health care
[61]	Craig et al. (2021)	Quantitative—survey (purposive sample) To describe the preliminary efficacy of AFFIRM Online	Guided cognitive behavior therapy (CBT)-based intervention DMH platform (AFFIRM Online)—a DMHI applying ICBT targeting LGBTQA+ youth (*n* = 46)	Feasibility and acceptability—effectiveness in the community-based implementation of AFFIRM Online for depression and coping with stress	Blended mental health care and waitlist control
[62]	Johansson et al. (2019)	RCT To determine the effectiveness of using the Swedish health care system’s ICBT platform	Guided CBT-based DMH platform (Swedish health care system)—targeting depression in routine psychiatry for adult patients (*n* = 108) with a primary diagnosis of major depressive disorder and excluding those with postpartum onset, ongoing alcohol- or substance abuse disorder, being assessed as high-risk suicidal patient, being actively engaging in self-harm, having a current eating disorder, bipolar disorder, ongoing psychotic symptoms, or co-occurring psychotherapy	Feasibility—preliminary evidence of efficacy for the Swedish health care system’s ICBT platform for treating depression in routine psychiatric care. Although there was a small study size and patients received general psychiatric care after the ICBT treatment which limits the implications	Blended mental health care and waitlist control
[63]	LaMonica et al. (2022)	Qualitative—focus group To describe 1) the codesign process of Innowell, 2) the DMH platform’s acceptance by stakeholders, and 3) evaluation to determine its impact at the level of the service user, health professional, and service	Integrated DMH platform’s (Innowell) performance indicators evaluated by representatives of stakeholders (i.e., Open Arms and headspace) for young people, Veteran and general population mental health care services (*n* = 84)	Feasibility—stakeholders support digital health in mental health care settings and simulations of Innowell for idealized implementation conditions are promising. Although organizational readiness for change, local-level leadership, appropriateness for end users and funding models hinder integration	Blended mental health care alone
[64]	Marcelle et al. (2019)	Quantitative—questionnaire To investigate the preliminary effectiveness of BetterHelp for providing psychotherapy	Multimodal psychotherapy DMH platform (BetterHelp)—active users self-reported on depression symptoms (*n* = 318)	Effectiveness—preliminary evidence of the use of BetterHelp in the treatment of adult depression. However, experimental trials are needed	Blended mental health care and usual primary care
[65]	O’Dea et al. (2021)	Mixed methods To evaluate the effectiveness of Smooth Sailing for help-seeking in students	Integrated DMH platform (Smooth Sailing) pilot trial—secondary students’ symptoms of anxiety and depression were screened and linked to online self-help or in-person care with a school counselor. Parents (*n* = 6) and school counselors (*n* = 4) were interviewed for their experiences with the delivery of the Smooth Sailing service model	Feasibility, acceptability and engagement—initial support for the use of Smooth Sailing in secondary schools to identify at-risk students. Benefits include ease of DMH platform use and psychoeducation. Although it requires parental consent, a higher uptake and engagement through frequent screening as well as targeting older students	Stepped mental health care and self-guided mental health care
[66]	Smit et al. (2021)	Qualitative–semi-structured interviews (purposive sample) To capture the user perspective on Depression Connect	Integrated DMH platform (Depression Connect)—experiences with an online peer support for individuals with depression (*n* = 15)—thematic analysis	Usability and engagement—the sample of users reported the peer support DMH platform is an accessible, safe and valuable tool to share depression coping experience. However, longitudinal research is required	Blended mental health care alone
[67]	Venning et al. (2021)	Qualitative—semi-structured interviews and focus groups To determine what people generally thought about the look, feel, and functionality of the DMH platform	Guided CBT-based (Low Intensity Virtual Coach) DMH Platform—experiences and engagement of a convenient sample of university students (*n* = 16) and mental health professionals (*n* = 5)	Acceptability and engagement—mostly negative experiences were reported indicating that the Virtual Coach was unrelatable and hard to engage with. The effectiveness of Virtual Coach DMH platforms appears to be limited due to low levels of acceptability and engagement	Blended mental health care alone
[68]	Vichta et al. (2018)	Mixed methods To facilitate young people’s perspective on the use and experiences of DMH platforms	An unspecified range of existing DMH platforms—interactive workshops and an online survey gathered young people’s (*n* = 404) perspectives on DMH platform integration into youth mental health care	Feasibility—DMH platforms can assist evaluating youth wellbeing over time. Although innovative approaches are required to gain qualitative data in a way that reaches young people in their own world	Blended mental health care alone

**Table 2 ijerph-20-00362-t002:** Overview of empirical studies that focused on evaluation of the DMHI/s applied on the DMH platform/s.

Ref-ere-nce	Authors	Study Design/Main Aim	DMH Platform (Type, Purpose of Use and Population)	Outcomes/Form of Evidence	Approach/Comparison
[69]	Bucci et al. (2018)	Qualitative—semi-structured interviews (purposive sample) To assess the feasibility and acceptability of Actissist, a digital health intervention	Guided CBT-based DMH platform intervention (Actissist) targeting youth psychosis—early psychosis service user (*n* = 21) perspectives	Feasibility and acceptability—largely positive views on the use of DMHIs for health care delivery. Although there are concerns over privacy and data security	Blended mental health care alone
[70]	Doukani et al. (2020)	Qualitative—semi-structured interviews (purposive sample) To examine the working alliance demands and adapt a conceptual framework to an intervention for depression	Guided CBT-based DMH platform intervention as part of E-compared trial—interviews of people with major depressive disorder (*n* = 19) to investigate design of the working alliance	Feasibility, usability and engagement—study is the first to offer a preliminary conceptual framework of the working alliance in ICBT for depression including how to establish, plan and promote a user-practitioner relationship in engagement strategies for technological design and clinical practice delivery	Blended mental health care alone
[71]	Goldkind and Wolf (2021)	Qualitative—interviews (purposive sample) To ask practitioners to describe their lived experience of providing tele-mental health services	Direct to consumer tele-mental health (DTCTMH) platforms (unspecified)—affordances of social work practitioners (*n* = 21)	Usability and engagement—key affordances of DTCTMH platforms include accessibility, anonymity, meaningful work, autonomy, lifelong learning, and access by new populations. Although there are hindering ethical complexities and structural challenges	Blended mental health care alone
[72]	Gray et al. (2020)	Qualitative—semi-structured interviews To elicit participant views on using SMART Recovery for routine outcome monitoring as a standard component of a mutual support group	Self-guided and guided DMH platform (SMART Recovery) for routine outcome monitoring, i.e., mutual support in addiction recovery—adults primarily with alcohol, drug and gambling addictions or other addictions (*n* = 20)	Feasibility—the use of SMART Recovery may complement physical, weekly group meetings. Although its use could pose a threat to in-person mutual support especially in cases with previous experience of such	Self-guided and guided mental health care
[73]	Hentati et al. (2021)	RCT To investigate differences in treatment engagement between two different user interfaces (UIs) for DMH services	Self-guided mental health problem-solving intervention DMH platform (Swedish health care system)—optimized UI versus basic UI DMH platform for the Swedish general population (*n* = 397)	Usability and engagement –optimized UI based on user experience (UX) design principles add to treatment engagement with the DMH platform, i.e., generating more solutions to behavioral problems. Although, the self-rated usability and treatment credibility may not be affected by whether the UI is optimized or not	Self-guided mental health care alone
[74]	Knapp et al. (2021)	Qualitative—focus groups To understand how digital tools can be integrated into settings that serve young people	Integrated DMH platform (centralized DMH platform to connect the clinician, young person, and young person’s family)—clinician perspectives (*n* = 37) on a desired integrated DMH platform to deliver mental health care for children and adolescents	Feasibility—Clinicians use digital tools to increase engagement and help young people build skills, facilitate learning, and monitor symptoms. However, a centralized DMH platform is recommended to improve accessibility by securely connecting the clinician, young person, and caregivers. Tailored solutions are required to serve youth-oriented needs	Blended mental health care alone
[75]	LaMonica et al. (2020)	Mixed methods To systematically monitor and evaluate the impact of implementing the InnoWell DMH Platform, into Australian mental health services to facilitate its refinement and the associated service model	Integrated DMH platform (Innowell)—evaluation of Project Synergy’s impact—surveys (*n* = 47), semi-stuctured interviews (*n* = 3), and workshops with representatives from health and social policy agencies, nongovernment organizations, primary care providers, emergency services, research institutions, community groups, and people with lived experience of suicide	Feasibility, acceptability and engagement—consensus that Innowell may benefit consumers and services. Although, implementation is hindered by a lack of readiness for change, e.g., technological infrastructure, digital literacy of staff and organizing who is responsible for recommending digital solutions	Blended mental health care alone
[76]	Richards et al. (2020)	RCT To evaluate the (cost-) effectiveness of ICBT for depression and anxiety in a pragmatic clinical trial within routine stepped care	Integrated-multifunctional DMH platform (SilverCloud)—ICBT for people with anxiety and depression disorders (*n* = 361), i.e., Improving Access to Psychological Therapies (IAPT) program	Effectiveness—SilverCloud’s ICBT is effective in >50% of people diagnosed with anxiety and/or depression (recovered after three months), cost-effective for IAPT after 12 months	Stepped mental health care and waitlist control
[77]	Sindoni et al. (2020)	Qualitative—case studies To provide analyses on how identity and distance of participants are indexed by focusing on how interpersonal relations are mapped linguistically and multimodally in #YouCanTalk on the Beyond Blue DMH platform.	Integrated DMH platform (Beyond Blue) applied in a case study on multimodal discourse analysis of peer support and professional mental health care for general populations targeting anxiety, depression and suicidality. A second case study on multimodal discourse analysis was applied with the #YouCanTalk web-based social media campaign and online support forum	Usability and engagement—the Beyond Blue DMH platform used direct language appropriate to target anxiety, depression and suicidality. #YouCanTalk is multimodal in terms of language, layout, modularity and content distribution, as well as pictures, infographics and videos. Although, more datasets are required to help understand how to reduce distance in mental health communication	Blended mental health care alone
[78]	Titov et al. (2020)	Quantitative—observational study To provide a summary of demographic characteristics and treatment outcomes for patients registered with MindSpot over its first 7 years of operation, including service use and symptom severity, and examined trends in these characteristics over time	Integrated-multifunctional DMH platform (MindSpot)—descriptive analysis of patients’ depression, anxiety and general distress and disability symptoms as well as post-treatment satisfaction (*n* = 121,652 screening users and 14,503 treatment users during a 7-year study)	Usability and engagement—a high assessment completion rate (78.9%); a very high rate (96.65%) of satisfaction with the MindSpot DMH platform; overall improvement in psychological symptoms sustained for 3 months after treatment; utility for a high volume DMH service. Although the relatively small size of registered sample limits generalizability	Self-guided and guided mental health care
[79]	Valentine et al. (2020)	Qualitative—semi-structured interviews To gain young people’s perspectives on the design and operation of a blended model of care in first-episode psychosis treatment	Integrated-multifunctional DMH platform (Horyzons, a derivative of MOST)—young people in first-episode psychosis treatment (*n* = 10)—perspectives on design and implementation	Acceptability—young people supported blended mental health care provided it assists face-to-face treatment. Although further research is needed on efficacy of the blended care approach by evaluating impact on the therapeutic alliance, clinical and social outcomes, cost-effectiveness, and engagement	Blended mental health care alone

## Data Availability

No new data were created or analyzed in this study. Data sharing is not applicable to this article.

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
