# Peer review of "Evaluation of the Use of Digital Mental Health Platforms and Interventions: Scoping Review"

_ijerph, 2022, doi:10.3390/ijerph20010362_

Round 1

Reviewer 1 Report

This topic is interesting. But there are some issues should be addressed.

The Abstract section is too long to focus, especially for the methods and result. And there are many redundant contents. For example, “The first author conducted screening, data extraction and full-text review…” I don’t think it’s necessary. It’s better to make it concise and clear.

Reviewer 2 Report

In this scoping review, the authors have reviewed the evaluation of digital platforms and digital mental health interventions. The overall scope of the paper is good because it summarizes existing digital platforms and tools and lets scholars perform a systematic review of the digital platforms for mental health care. This research topic is relevant to the general audience of this journal. I believe this review adds value to present scientific knowledge but has some major drawbacks. Accordingly the manscript cannot be accepted in its current form.

Major comments

• Although the terms digital platforms, DMH, etc. are defined at the end of the paper, it is recommended to differentiate and describe digital platforms, DMH, DMHIs, and DMH platforms in the introduction section to let readers know what the definition of each term is.

• Although RCTs are different from survey study designs, they still fall under the umbrella of quantitative studies. Consequently, I am not sure that the significant number of studies were qualitative in nature.

• Another concern is that it appears the authors do not distinguish between different study designs as they are defined in experimental psychology and research methods. Consequently, it is suggested to review Denny, E., & Weckesser, A. (2018). Qualitative research: what it is and what it is not. BJOG: An International Journal of Obstetrics and Gynaecology and revise accordingly.

Minor comments

• Lines 47-48: Why is the conundrum (complex relationship between mental health and suicide) exacerbated by integration of telehealth? This sentence does not follow the former

• Line: 62: Again, what is the difference between digital platform, DMH, and DMHI?

• Line: 71: The heading “use of DMH..” is not specific. If it is functionality discussed then that should be in the subheading.  And what is meant by functionality? An example was given, but it is not defined. It would be good to define functionality clearly to the reader and only then give examples and later effectiveness.

• Line 95: There is inconsistency regarding the research question: What is being addressed?  Mental health? Depression? Suicide? It seems to include all of the above. If that is the case than this should be addressed and the reasons for including all of these mental health disorders should be explained.

• Line 105: RCT should be defined. This is done in the methods section in lines 180- 182 but should appear before.

• Line 114: not a sentence; same for lines 286-289.

• In line 222, the word “and” in “…Google Scholar and were chosen to include …” should be removed.

• There is no caption for Figure 1

• Line 448: what is “ascending and descending order”?
